# Kinetic mechanism of Na$^+$-coupled aspartate transport catalyzed by Glt$_{Tk}$

Gianluca Trinco[1], Valentina Arkhipova[1,4], Alisa A. Garaeva[1,5], Cedric A. J. Hutter [2], Markus A. Seeger [2], Albert Guskov [1,3] & Dirk J. Slotboom [1✉]

It is well-established that the secondary active transporters Glt$_{Tk}$ and Glt$_{Ph}$ catalyze coupled uptake of aspartate and three sodium ions, but insight in the kinetic mechanism of transport is fragmentary. Here, we systematically measured aspartate uptake rates in proteoliposomes containing purified Glt$_{Tk}$, and derived the rate equation for a mechanism in which two sodium ions bind before and another after aspartate. Re-analysis of existing data on Glt$_{Ph}$ using this equation allowed for determination of the turnover number (0.14 s$^{-1}$), without the need for error-prone protein quantification. To overcome the complication that purified transporters may adopt right-side-out or inside-out membrane orientations upon reconstitution, thereby confounding the kinetic analysis, we employed a rapid method using synthetic nanobodies to inactivate one population. Oppositely oriented Glt$_{Tk}$ proteins showed the same transport kinetics, consistent with the use of an identical gating element on both sides of the membrane. Our work underlines the value of bona fide transport experiments to reveal mechanistic features of Na$^+$-aspartate symport that cannot be observed in detergent solution. Combined with previous pre-equilibrium binding studies, a full kinetic mechanism of structurally characterized aspartate transporters of the SLC1A family is now emerging.

[1] Groningen Biomolecular Sciences and Biotechnology Institute, University of Groningen, Groningen, The Netherlands. [2] Institute of Medical Microbiology, University of Zurich, Zurich, Switzerland. [3] Moscow Institute of Physics and Technology, Dolgoprudny, Russia. [4] Present address: ZoBio BV, Leiden, The Netherlands. [5] Present address: Institute of Medical Microbiology, University of Zurich, Zurich, Switzerland. ✉email: d.j.slotboom@rug.nl

Excitatory amino acid transporters (EAATs) of the solute carrier family 1A (SLC1A) take up the neurotransmitter L-glutamate from the synaptic environment, which is necessary to keep the extracellular concentration low and prevent neurotoxicity[1,2]. EAATs couple uptake of one amino acid substrate molecule to the co-transport of three sodium ions and one proton and counter-transport of one potassium ion[3–6]. Thus, glutamate gradients of a million-fold across the membrane under resting conditions can be sustained. The closely related archaeal transporters Glt$_{Ph}$ from *Pyrococcus horikoshii* and Glt$_{Tk}$ from *Thermococcus kodakarensis* of the SLC1A family (78% sequence identity to each other, ~36% sequence identity to EAATs) take up aspartate rather than glutamate in symport with three sodium ions and are not coupled to potassium or proton transport[7–11]. These prokaryotic homologs of the neurotransmitter transporters have been instrumental in delineating shared structural features of this transporter family[7–9,12–22].

SLC1A family proteins are homotrimers, with independently operating protomers[19,21,23–29], each organized in two domains. A rigid scaffold domain mediates all the contacts with the neighboring protomers, and a peripheral transport domain binds the amino acid substrate and cations[13,16,30,31]. The transport domains are mobile and move through the lipid bilayer (alike an elevator) when translocating the amino acid substrate and co-transported ions across the membrane[13]. During movement of the transport domain, the substrate-binding site is occluded from the solvent and shielded by the tips of two pseudo-symmetrical helical hairpins (HP1 and HP2). The latter hairpin is a gating element that can hinge between a closed position (taken during elevator movements) and an open position (allowing loading or release of the substrate and co-transported ions). The extent of the elevator-like movement of the transport domain is so large (~20 Å in Glt$_{Tk}$) that HP2 acts as a gating element both on the extracellular and the intracellular side of the membrane.

Transport assays in proteoliposomes have revealed that both Glt$_{Ph}$ and Glt$_{Tk}$ catalyze electrogenic transport with a strict stoichiometry of three co-transported Na$^+$ ions per aspartate[14,32]. Data from studies on equilibrium binding and pre-equilibrium kinetics of binding with the solubilized proteins in detergent solution have shown that co-transported ions and aspartate bind in a highly cooperative way, which is crucial to ensure thermodynamic coupling[10,15,33–37]. These experiments have indicated that most likely two sodium ions bind first, then aspartate, and finally the third sodium ion. The binding of the last Na$^+$ leads to gate closure of HP2, which in turn is a prerequisite for elevator-like conformational changes that translocate the bound cargo across the membrane. Structures of Glt$_{Ph}$ and Glt$_{Tk}$ have provided a qualitative explanation for the observed binding order. Two of the sodium binding sites (named Na1 and Na3) are buried deep in the proteins[15]. A substantial conformation rearrangement in the apo-protein (most pronounced in the conserved unwound region of TM7) is required to create the geometry needed for sodium binding, which makes this step slow. The conformational rearrangement, which is stabilized by the binding of the two sodium ions, also affects residues involved in aspartate binding. While the apo-state does not have a measurable affinity for aspartate, sodium binding creates a high-affinity site for the amino acid substrate. The last sodium ion binds to a site in direct contact with the HP2 gate (Na2), and locks the gate in the closed position, with aspartate and the three sodium ions occluded from the environment.

Despite the enormous amount of data on structure and binding mechanism in detergent solution, insight into the kinetic mechanism under translocating conditions is fragmented and incomplete. Here, we set out to determine the kinetic mechanism of aspartate transport by using purified Glt$_{Tk}$, reconstituted in proteoliposomes. We measured initial rates of transport at a wide range of substrate and co-ion concentrations, a method that has been used extensively for the mechanistic characterization of enzymes, leading to insight into the order of binding of substrates[38]. This method has not been used much on purified membrane transporters, in part because it is often impossible to control the orientation of the reconstituted transporters in proteoliposomes, leading to mixed populations, thus complicating kinetic analysis.

To overcome this latter problem, we isolated an inhibitory synthetic nanobody[39], which we used to inactivate membrane transporters oriented in one of the two possible orientations. This method is rapid, generally applicable, and does not depend on mutagenesis and chemical modifications. The work presented here, combined with the results of previous pre-equilibrium binding studies[36], allows for the determination of accurate turnover numbers, which we illustrate by analyzing available data for aspartate transport by Glt$_{Ph}$.

## Results

To study the kinetic mechanism of Na$^+$-coupled aspartate transport by Glt$_{Tk}$, we used a classical enzymology method, in which we measured the initial uptake rates of radiolabelled L-aspartate into proteoliposomes reconstituted with purified Glt$_{Tk}$, as a function of the external concentrations of L-aspartate and Na$^+$. To ensure initial rate conditions, aspartate and Na$^+$ were absent from the lumen of the liposomes at the onset of the experiment, and the rate was determined from the linear part of the uptake experiment (Supplementary Fig. 1). In the first set of experiments (presented in Figs. 1–3 and Tables 1–3), we measured the combined transport activity of proteins with right-side-out and inside-out membrane orientation in the liposomes, as we did not inactivate either of the two populations. These experiments can be compared with binding experiments performed in detergent-solution where the sidedness is absent. In follow-up experiments (presented in Fig. 4 and Table 4), we silenced one of the two populations, which allows for comparison with experiments in which the transporters had been fixed in a single orientation by crosslinking[35,36].

We managed to determine accurate transport rates using external Na$^+$ concentrations in the range between 5 and 300 mM and aspartate concentrations between 50 nM and 100 μM (Table 1). The upper and lower boundaries of the concentration ranges were set by practical considerations. Aspartate concentrations higher than 100 μM required large dilution of the radiolabelled amino acid with unlabeled aspartate, which caused poor signal-to-noise levels in the uptake experiments. Na$^+$ concentrations higher than 300 mM could not be used, because the preparation of proteoliposomes in buffer containing high salt concentrations prevented the formation of a firm pellet when centrifuged, therefore making it impossible to reach the necessary protein concentration for the experiments. In the low concentration regime, conditions in which 1 mM Na$^+$ was used in combination with aspartate concentrations lower than 1 μM resulted in poor signal-to-noise ratios. Despite these limitations, the range of concentrations was sufficient to provide insight into the kinetic mechanism.

The results of the uptake experiments in liposomes with mixed protein orientations are summarized in Table 1, where each row contains the initial rates of transport ($v_0$) at a fixed sodium concentration, but with increasing aspartate concentrations. When analyzing $v_0$ as a function of the aspartate concentration row-by-row, we found that rectangular hyperbolic functions fitted the data well (Fig. 1a), which allowed for the determination of the apparent maximal rates of transport ($v_{max}^{Asp}$ (app)) and apparent Michaelis-Menten constants ($K_M^{Asp}$ (app)) (Table 2). The superscript "Asp" indicates that the aspartate was varied, while the sodium concentration was kept constant (hence "apparent").

Each column in Table 1 contains the measured initial rates of aspartate transport at increasing Na$^+$ concentrations while

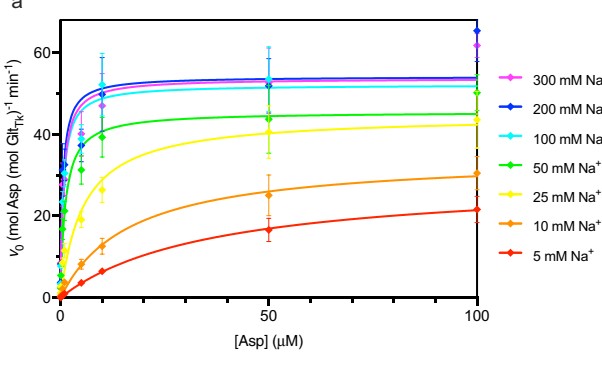

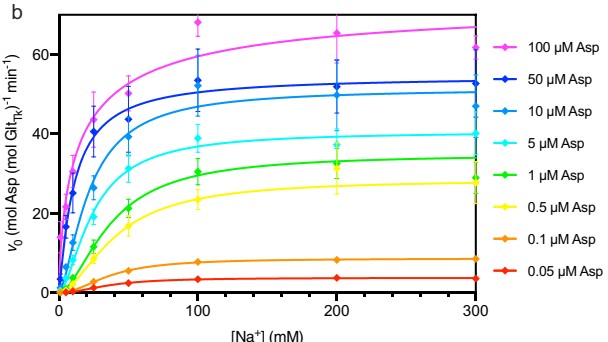

**Fig. 1 L-Asp transport rates catalyzed by purified and reconstituted Glt$_{Tk}$ as a function of the concentrations of Na$^+$ and L-aspartate.** The rates represent the combined contributions of right-side-out and inside-out oriented proteins. **a** Aspartate-dependent measurements at different fixed Na$^+$ concentrations. The lines represent fits of the Michaelis-Menten equation to the data for uptake at Na$^+$ concentrations of 5 mM (red), 10 mM (orange), 25 mM (yellow), 50 mM (green), 100 mM (cyan), 200 mM (blue), 300 mM (purple). **b** Sodium-dependence of transport at fixed L-Asp concentrations. The lines represent fits of the Hill equation to the data for uptake at 0.05 μM (red), 0.1 μM (orange), 0.5 μM (yellow), 1 μM (green), 5 μM (cyan), 10 μM (light blue), 50 μM (blue), 100 μM (purple). Each uptake rate represents the average of three independent biological replicates, each constituted by two technical replicates, and the standard error of the mean is shown.

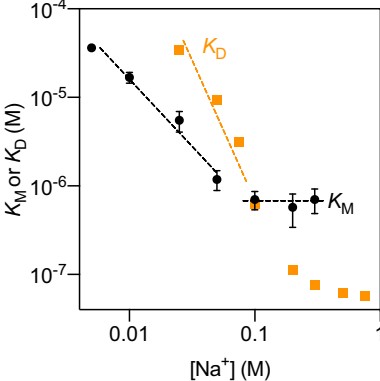

**Fig. 2 Dependence of the apparent affinities for L-Asp ($K_M^{Asp}$ (app)) on the sodium ion concentration (black symbols).** Dashed lines represent linear fits to data points in the low and high sodium ion concentration regimes. Each value represents the average of three independent biological replicates, and two technical replicates. The standard error of the mean is shown. For comparison, the dissociation constants $K_D^{Asp}$(app) determined previously are also plotted (orange)[15]. The data represent the combined contributions of right-side-out and inside-out oriented proteins.

maintaining a fixed aspartate concentration. Analysis of these rates as a function of the Na$^+$ concentration revealed sigmoidal dependencies (Fig. 1b). The Hill equation was used to fit the data, yielding values for $v_{max}^{Na}$ (app) and $K_M^{Na}$ (app) and the apparent Hill coefficient $n_{Hill}$ (app) (Table 3). In this case, the superscript "Na" indicates sodium-dependent measurements, and "apparent" indicates that the measurements were done at a fixed L-aspartate concentration.

For interpretation of the uptake data in the framework of a kinetic model of transport, the apparent affinities for L-aspartate ($K_M^{Asp}$ (app)) and the apparent maximal rates ($v_{max}^{Asp}$ (app) and $v_{max}^{Na}$ (app)) are the most informative parameters. $K_M^{Na}$ (app) and $n_{Hill}$ (app) contain less useful information for discrimination between different mechanisms (as discussed in references [38,40]), and therefore these values were not used further here.

**$K_M$ analysis reveals a complex mechanism.** Information on the kinetic mechanism is contained both in the dependence of the apparent affinity for aspartate $K_M^{Asp}$ (app) on the Na$^+$ concentration and in potential differences between $K_M^{Asp}$ (app) and the equilibrium constant for L-aspartate binding ($K_D^{Asp}$ (app)). Table 2 shows that $K_M^{Asp}$ (app) was strongly dependent on the Na$^+$ concentration at concentrations below 50 mM Na$^+$. At Na$^+$ concentrations above 100 mM, $K_M^{Asp}$ (app) became independent of the Na$^+$ concentration and leveled off to 0.7 μM (Table 2), a value that is important for mechanistic interpretation (discussed in detail below).

Comparison of the $K_M^{Asp}$ (app) values with the equilibrium constants for L-aspartate binding ($K_D^{Asp}$ (app)), determined previously by isothermal titration calorimetry (ITC) in detergent solution[15], revealed almost identical values at a Na$^+$ concentration of 100 mM, but large differences at higher and lower concentrations of Na$^+$ (Fig. 2). While $K_M^{Asp}$ (app) was an order of magnitude higher than the equilibrium constant for binding at a sodium concentration of 300 mM ($7.0 \times 10^2$ nM versus 75 nM), at concentrations below ~75 mM, $K_M^{Asp}$ (app) was up to an order of magnitude lower than the $K_D^{Asp}$ (app). Such discrepancies are indicative of a complex kinetic mechanism that cannot be interpreted in the conceptual framework of the rapid equilibrium approximation, which is based on the assumption that the transport step (described by turnover number $k_{cat}$) is much slower than the establishment of the binding equilibrium of sodium ions and aspartate, described by the equilibrium constants $K_D$. The rapid equilibrium assumption was previously also dismissed for aspartate transport by Glt$_{Ph}$, based on a more limited comparison of $K_M$ and $K_D$[10], and on pre-equilibrium binding experiments in detergent solution[36], but the data presented here, based on the comparison of transport and binding experiments over a broad range of sodium concentrations, revealed a variable ratio between $K_M^{Asp}$ (app) and $K_D^{Asp}$ (app) depending on the Na$^+$ concentration, which is indicative of kinetic complexity. It is noteworthy that the rapid equilibrium assumption might hold at very low Na$^+$ concentration[36], but as discussed above, the sensitivity of the radiolabel-based transport assays is not high enough to measure aspartate uptake in such conditions.

**Mechanistic interpretation of transport data using the steady-state approximation.** Because the rapid equilibrium approximation was found invalid, we turned to analysis based on the steady-state assumption. While the Michaelis–Menten or Hill equations can always describe the substrate dependencies of the uptake rates when the rapid equilibrium approximation is valid, it is possible to find more complex relations in the steady-state analysis, depending on the details of the kinetic mechanism[38]. For instance, for some kinetic mechanisms, $v_{max}$ values may be (local)

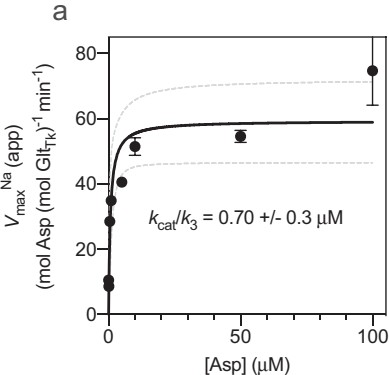
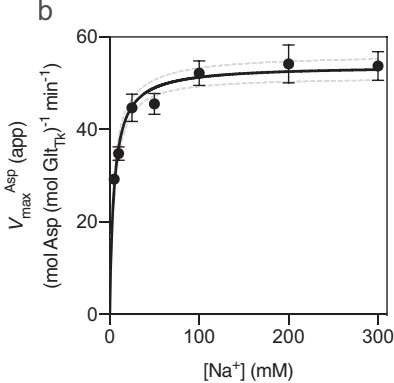

**Fig. 3 Dependence of the maximal rates L-Asp transport rate on the concentrations of Na$^+$ and L-aspartate. a** Dependence of the maximal rates of transport $v_{max}^{Na}$ (app) from Fig. 1a on the concentration of L-Asp. **b** Dependence of the maximal rates of transport $v_{max}^{Asp}$ (app) from Fig. 1b on the concentration of sodium ions. Solid and dashed lines represent fits of rectangular hyperbolic functions to the data and the 95% confidence intervals, respectively. In panel A the fitted value of $k_{cat}/k_3$ (Eq. (13)) is indicated. Each value represents the average of three independent biological replicates, and two technical replicates. The standard error of the mean is shown. The data represent the combined contributions of right-side-out and inside-out oriented proteins.

**Table 1 Initial rates of L-aspartate uptake by Glt$_{Tk}$ reconstituted in proteoliposomes. In the first column, the concentrations in parentheses indicate the amount of choline chloride used in the external reaction buffer to balance the osmotic and ionic strength. Each uptake rate represents the average of three independent biological replicates and independent reconstitutions, each with two technical replicates. The standard error of the mean is indicated.**

| [L-Asp]→ [Na$^+$]↓ | 0.05 µM | 0.1 µM | 0.5 µM | 1 µM | 5 µM | 10 µM | 50 µM | 100 µM |
|---|---|---|---|---|---|---|---|---|
| 1 mM (299 mM) | Not determined | Not determined | Not determined | 0.03 ± 2.6 10$^{-2}$ min$^{-1}$ | 0.51 ± 0.3 min$^{-1}$ | 0.25 ± 0.3 min$^{-1}$ | 2.85 ± 1.2 min$^{-1}$ | 10.93 ± 4.4 min$^{-1}$ |
| 5 mM (295 mM) | 0.05 ± 1.6 10$^{-2}$ min$^{-1}$ | 0.06 ± 2.1 10$^{-2}$ min$^{-1}$ | 0.59 ± 0.1 min$^{-1}$ | 0.96 ± 0.2 min$^{-1}$ | 3.62 ± 0.7 min$^{-1}$ | 6.48 ± 0.9 min$^{-1}$ | 16.59 ± 2.8 min$^{-1}$ | 21.62 ± 3.2 min$^{-1}$ |
| 10 mM (290 mM) | 0.21 ± 5.5 10$^{-2}$ min$^{-1}$ | 0.28 ± 0.1 min$^{-1}$ | 2.16 ± 0.4 min$^{-1}$ | 3.75 ± 0.6 min$^{-1}$ | 8.19 ± 1.2 min$^{-1}$ | 12.60 ± 1.9 min$^{-1}$ | 25.08 ± 5.0 min$^{-1}$ | 30.51 ± 4.1 min$^{-1}$ |
| 25 mM (275 mM) | 0.95 ± 0.2 min$^{-1}$ | 1.46 ± 0.5 min$^{-1}$ | 8.53 ± 1.2 min$^{-1}$ | 11.56 ± 1.7 min$^{-1}$ | 19.09 ± 2.0 min$^{-1}$ | 26.39 ± 3.0 min$^{-1}$ | 40.58 ± 6.4 min$^{-1}$ | 43.59 ± 6.9 min$^{-1}$ |
| 50 mM (250 mM) | 1.88 ± 0.4 min$^{-1}$ | 3.34 ± 1.0 min$^{-1}$ | 16.79 ± 2.6 min$^{-1}$ | 21.26 ± 2.3 min$^{-1}$ | 31.28 ± 3.5 min$^{-1}$ | 39.30 ± 4.9 min$^{-1}$ | 43.69 ± 8.3 min$^{-1}$ | 50.21 ± 4.4 min$^{-1}$ |
| 100 mM (200 mM) | 2.75 ± 0.5 min$^{-1}$ | 4.79 ± 1.5 min$^{-1}$ | 23.46 ± 2.5 min$^{-1}$ | 30.47 ± 3.3 min$^{-1}$ | 38.90 ± 3.5 min$^{-1}$ | 52.16 ± 7.7 min$^{-1}$ | 53.52 ± 7.9 min$^{-1}$ | 68.13 ± 3.5 min$^{-1}$ |
| 200 mM (100 mM) | 3.06 ± 0.6 min$^{-1}$ | 4.83 ± 1.5 min$^{-1}$ | 31.27 ± 6.4 min$^{-1}$ | 32.55 ± 3.8 min$^{-1}$ | 37.29 ± 4.0 min$^{-1}$ | 49.80 ± 9.0 min$^{-1}$ | 51.90 ± 6.7 min$^{-1}$ | 65.40 ± 7.5 min$^{-1}$ |
| 300 mM (0 mM) | 2.93 ± 0.7 min$^{-1}$ | 4.96 ± 1.6 min$^{-1}$ | 27.72 ± 5.3 min$^{-1}$ | 28.95 ± 4.0 min$^{-1}$ | 40.12 ± 5.4 min$^{-1}$ | 46.98 ± 7.9 min$^{-1}$ | 52.69 ± 8.5 min$^{-1}$ | 61.77 ± 2.9 min$^{-1}$ |

**Table 2 $v_{max}^{Asp}$ (app) and $K_M^{Asp}$ (app) values for aspartate dependent uptakes obtained at constant [Na$^+$].**

| [Na$^+$] | $v_{max}^{ASP}$ (app) (min$^{-1}$) | $K_M^{ASP}$ (app) (µM) |
|---|---|---|
| 5 mM | 29.2 ± 0.58 | 36 ± 1.9 |
| 10 mM | 34.8 ± 1.5 | 16.8 ± 2.4 |
| 25 mM | 44.7 ± 2.9 | 5.5 ± 1.5 |
| 50 mM | 45.5 ± 2.3 | 1.2 ± 0.3 |
| 100 mM | 52.1 ± 2.6 | 0.70 ± 0.16 |
| 200 mM | 54.2 ± 4.1 | 0.58 ± 0.23 |
| 300 mM | 53.7 ± 3.1 | 0.70 ± 0.22 |

Values are derived from the data presented in Table 1 and Fig. 1a. Each uptake rate represents the average of three independent biological replicates, each with two technical replicates. The standard error of the mean is indicated.

initial rates are described well by rectangular hyperbola and sigmoidal curves (Figs. 1 and 2), suggesting that such random steps do not play a significant role, at least not in the concentration regime that we used. This notion is consistent with the kinetic binding model derived for detergent-solubilized Glt$_{Ph}$, where the binding of sodium ions and aspartate to Glt$_{Ph}$ was found to be ordered in the concentration range between 5 and 300 mM Na$^+$, with two sodium ions binding before and one after aspartate[36]. This ordered binding mechanism is also consistent with a recent study that revealed a conformational selection step leading to the binding of the first sodium ion[34]. Therefore, we chose to analyze the transport data presented here with the following kinetic model:

$$E \underset{k_{-1}}{\overset{k_1[Na^+]}{\rightleftharpoons}} \{ENa\} \underset{k_{-2}}{\overset{k_2[Na^+]}{\rightleftharpoons}} \{ENaNa\} \underset{k_{-3}}{\overset{k_3[Asp]}{\rightleftharpoons}} \{ENaNaAsp\} \underset{k_{-4}}{\overset{k_4[Na^+]}{\rightleftharpoons}} \{ENaNaAspNa\} \overset{k_{cat}}{\rightarrow}$$

(1)

in which $E$ designates Glt$_{Tk}$.

While we base our further analysis on the mechanism shown in Eq. (1), we will show in the discussion section that the main conclusions hold for any mechanism in which at least one sodium ion binds after aspartate.

maxima, instead of being reached asymptotically, and $K_M$ values may be undefined. Such complex relations often arise when steps occur in the mechanism where two different substrates (corresponding to aspartate and Na$^+$ in the case of Glt$_{Tk}$) bind randomly[38]. In our data, the concentration dependences of the

**Table 3 $v_{max}^{Na}$ (app), $K_M^{Na}$ (app) and $n_{Hill}$ (app) values for Na⁺ dependent uptakes obtained at constant [Asp].**

| [L-Asp] | $v_{max}^{Na}$ (app) (min⁻¹) | $K_M^{Na}$ (app) (mM) | $n_{Hill}$ (app) |
|---|---|---|---|
| 0.05 μM | 3.7 ± 0.1 | 36.7 ± 1.8 | 2.1 0.19 |
| 0.1 μM | 8.6 ± 0.1 | 37.0 ± 0.81 | 2.0 ± 0.08 |
| 0.5 μM | 28.5 ± 0.15 | 40.9 ± 0.42 | 1.70 ± 0.025 |
| 1 μM | 34.9 ± 1.0 | 37.3 ± 2.0 | 1.7 ± 0.13 |
| 5 μM | 40.6 ± 1.3 | 24.7 ± 2.1 | 1.6 ± 0.19 |
| 10 μM | 51.4 ± 2.7 | 21.8 ± 3.2 | 1.5 ± 0.29 |
| 50 μM | 54.6 ± 1.9 | 10.9 ± 1.2 | 1.1 ± 0.14 |
| 100 μM | 72.34 ± 10.5 | 14.8 ± 7.9 | 0.72 ± 0.21 |

Values are derived from the data presented in Table 1 and Fig. 1b. Each uptake rate represents the average of three independent biological replicates, each with two technical replicates. The standard error of the mean is indicated.

**Table 4 $v_{max}^{Asp}$ (app) and $K_M^{Asp}$ (app) values for aspartate dependent uptakes obtained at constant [Na⁺] in the presence of 750 nM sybody on the outside of the liposomes.**

| [Na⁺] | $v_{max}^{Asp}$ (app) (min⁻¹) | $K_M^{Asp}$ (app) (μM) |
|---|---|---|
| 10 mM | 9.8 ± 1.0 | 4.9 ± 1.4 |
| 100 mM | 18.3 ± 2.0 | 0.97 ± 0.36 |
| 200 mM | 17.0 ± 1.0 | 0.65 ± 0.13 |
| 300 mM | 17.2 ± 1.5 | 0.63 ± 0.17 |

Each uptake rate represents the average of two independent biological replicates, each with two technical replicates. The standard error of the mean is indicated.

To derive a rate equation for this kinetic model of Eq. (1), we used the King-Altman method[38,41]:

$$\frac{v_0}{v_{max}} = \frac{a_1[Na]^3[Asp]}{a_1[Na]^3[Asp] + a_2[Na]^2[Asp] + a_3[Na][Asp] + a_4[Na]^3 + a_5[Na]^2 + a_6[Na] + a_7}$$ (2)

in which $v_{max}$ is the maximal attainable rate of transport at high Na⁺ and L-aspartate concentrations, and $a_1$–$a_7$ are expressions of rate constants:

$$a_1 = k_1 k_2 k_3 k_4$$ (3)

$$a_2 = k_2 k_3 k_4 k_{cat} + k_1 k_3 k_4 k_{cat} + k_1 k_2 k_3 k_{cat} + k_1 k_2 k_3 k_{-4}$$ (4)

$$a_3 = k_{-1} k_3 k_4 k_{cat}$$ (5)

$$a_4 = k_1 k_2 k_4 k_{cat}$$ (6)

$$a_5 = k_1 k_{-2} k_4 k_{cat} + k_1 k_2 k_{-3} k_{cat} + k_1 k_2 k_{-3} k_{-4}$$ (7)

$$a_6 = k_1 k_{-2} k_{-3} k_{cat} + k_{-1} k_{-2} k_4 k_{cat} + k_1 k_{-2} k_{-3} k_{-4}$$ (8)

$$a_7 = k_{-1} k_{-2} k_{-3} k_{cat} + k_{-1} k_{-2} k_{-3} k_{-4}$$ (9)

Equation (2) can be rearranged to derive expressions for $v_{max}^{Na}$ (app) and $v_{max}^{Asp}$ (app):

$$v_{max}^{Asp}(app) = v_{max} * \frac{a_1[Na]^2}{a_1[Na]^2 + a_2[Na] + a_3}$$ (10)

$$v_{max}^{Na}(app) = v_{max} * \frac{[Asp]}{\frac{a_4}{a_1} + [Asp]} = v_{max} * \frac{[Asp]}{\frac{k_{cat}}{k_3} + [Asp]}$$ (11)

Thus, the model predicts that both $v_{max}^{Na}$ (app) and $v_{max}^{Asp}$ (app) are dependent on the concentration of the other substrate, which is fully consistent with the data presented in Fig. 3a, b. Moreover, Eq. (11) describes a rectangular hyperbolic relation between $v_{max}^{Na}$ (app) and [Asp]. By fitting the Michaelis-Menten equation to the data (Fig. 3a), we found a value for $k_{cat}/k_3$ of 0.7 μM. Since the turnover number $k_{cat}$ is known from the $v_{max}$ data (~0.9 s⁻¹ (~54 min⁻¹), see Fig. 3, Table 2), a value of ~$1.3 \times 10^6$ M⁻¹ s⁻¹ for $k_3$ is derived, remarkably similar to the value of $1.2 \times 10^6$ M⁻¹ s⁻¹ that was found for Glt$_{Ph}$, obtained in pre-equilibrium binding experiments[36].

Equation (2) can also be rearranged to derive an expression for $K_M^{Asp}$ (app):

$$K_M^{Asp}(app) = \frac{a_4[Na]^3 + a_5[Na]^2 + a_6[Na] + a_7}{a_1[Na]^3 + a_2[Na]^2 + a_3[Na]}$$ (12)

From Eq. (12) the values for $K_M^{Asp}$ (app) that are reached in the low and high [Na⁺] regimes can be found:

$$\lim_{[Na]\to} K_M^{Asp}(app) = \frac{a_4}{a_1} = \frac{k_{cat}}{k_3}$$ (13)

$$\lim_{[Na]\to 0} K_M^{Asp}(app) = \frac{a_7}{a_3[Na]}$$ (14)

Equation (13) predicts that in the high concentration limit a constant value for $K_M^{Asp}$ (app) is reached, which equals the ratio between two rate constants $k_{cat}/k_3$. The data presented in Fig. 2 and Table 2 show that the value of $K_M^{Asp}$ (app) levels off to ~0.7 μM at high Na⁺ concentration. Since the turnover number $k_{cat}$ is ~0.9 s⁻¹ (Fig. 3 and Table 2), a value of ~$1.3 \times 10^6$ M⁻¹ s⁻¹ for $k_3$ is found. Thus, two approaches (analysis of $v_{max}^{Na}$ (app) and $K_M^{Asp}$ (app)), reveal a value for $k_3$ that agrees well with the value of $1.2 \times 10^6$ M⁻¹ s⁻¹ that was found for Glt$_{Ph}$.

When the apparent affinity constants are plotted in a double logarithmic plot against the concentration of Na⁺, linear relations are approached in both the high and low Na⁺ concentration extremes (Fig. 2). The slope is zero at high Na⁺ concentration (because $K_M^{Asp}$ (app) levels off to $k_{cat}/k_3$), and a slope of −1.4 is found in the low Na⁺ concentration regime, which deviates from the slope of −1 predicted by the model. This discrepancy may indicate that lower concentrations of sodium should have been used to meet the conditions for Eq. (14) to be valid, something which was impossible for technical reasons, as discussed above. Alternatively, the deviation might be caused by the experimental error inherent to the transport measurements at low Na⁺ concentrations. The slope of the log($K_M$) plot in the low Na⁺ concentration regime is approximately twofold shallower than that of the log($K_D$) plot, which is consistent with the binding model[36].

**Side specific inhibition with sybody**. Although the analysis presented above is internally consistent, as well as consistent with existing kinetic data for binding on $Glt_{Ph}$, there is a potential complication caused by the proteoliposome system used, because the reconstitution procedure usually results in a mix of inside-out- and right-side-out-oriented proteins in the bilayer. For instance, it has been demonstrated that $Glt_{Ph}$ reconstitutes in the two orientations with equal probability[10]. If the oppositely oriented proteins take up aspartate via different kinetic mechanisms (the equivalence of different mechanisms for forward and reverse transport in vivo), the results of the kinetic analysis could be convoluted and potentially lead to misinterpretation.

To determine to which extent the kinetic mechanism depends on the orientation of $Glt_{Tk}$ in liposomes, we set out to inactivate either the right-side-out or the inside-out oriented transporters. Inhibition of the transporter from only one side of the membrane by modification of cysteine mutants did not work for $Glt_{Tk}$, possibly because of the one-gate nature of the elevator mechanism, in which the identical binding site residues and gating elements are alternately exposed to either side of the membrane[13]. Therefore, we chose to explore an alternative method by using synthetic nanobodies (sybodies)[39,42]. Since sybodies recognize water-exposed surface epitopes and are membrane-impermeable, they are expected to be suitable for orientation-specific inhibition. We selected 42 unique sybodies against $Glt_{Tk}$, using an established platform, which included ribosome display, two rounds of phage display, and ELISA[39].

One of these sybodies (sybody 1) completely blocked aspartate transport by $Glt_{Tk}$ when added from both sides of the membrane (in the lumen and in the external solution), but inhibited partially when added only on the outside of the proteoliposomes (Fig. 4a). It is important to note that the procedure to load sybodies in the liposome lumen includes an extrusion step in which all $Glt_{Tk}$ molecules in the sample are exposed to the sybody. Therefore, it is not possible to do the opposite experiment, with the sybody exclusively included in the liposome lumen. Regardless of this limitation, the experiment conclusively showed that the sybody causes the sought-after sidedness of inhibition, and the result suggests that $Glt_{Tk}$ had reconstituted in both orientations in the proteoliposomes, similar to what was shown for $Glt_{Ph}$ before. To explain the inhibitory properties of sybody 1, we solved a structure of the sybody-$Glt_{Tk}$ complex using single-particle cryo-EM. The sybody binds on the extracellular surface of $Glt_{Tk}$ at the interface between the transport and scaffold domain. The bound sybody thereby makes the elevator movement impossible, which prevents transport (Fig. 4b). The sybody thus inactivates the $Glt_{Tk}$ molecules with right-side-out orientation, and the residual uptake activity can be attributed to the inside-out oriented proteins.

We chose to use this sybody, added only on the outside of the proteoliposomes, to repeat a subset of the experiments described above. First, we tested whether $v_{max}^{Na}$ (app) still depended on the aspartate concentration and whether $v_{max}^{Asp}$ (app) still depended on the $Na^+$ concentration when the right-side-out oriented molecules were inactivated by the sybody. Indeed, both $v_{max}$ (app) values still varied with the concentrations of the co-substrate (Fig. 4c, d), consistent with the kinetic mechanism (Eqs. (10) and (11)). Second, we determined whether a constant value for $K_M^{Asp}$ (app) was still reached in the high $Na^+$ concentration limit, as predicted by Eq. (13). Indeed, a constant value of ~0.6 μM was found above 100 mM $Na^+$, which intriguingly did not deviate significantly from the value in the dual-population proteoliposomes (Table 5). Assuming that the $k_{cat}$ value of the active population of inside-out oriented proteins was still ~0.9 s$^{-1}$, the value of $k_3$ remained unaltered. In other words, the measurable

parameters of the kinetic mechanism of the right-side-out and inside-out oriented proteins were very similar.

## Discussion

Reconstitution of purified membrane proteins in liposomes often leads to mixed-orientation in lipid bilayers. For secondary active transporters, which can readily operate in both directions, the co-existence of right-side-out and inside-out oriented proteins is problematic for kinetic analysis. The work presented here shows that inactivation of transporter from one side of the membrane using a synthetic nanobody (sybody) is an effective way to deal with a mixed-orientation upon reconstitution in liposomes[39,42]. Synthetic nanobodies are membrane impermeable, and highly specific for the binding epitope. While also natural nanobodies could be used for this purpose, sybodies offer a major advantage, because the selection can be carried out under defined buffer conditions, which may be used to steer the selection towards binders of a specific state. Also, the immobilization method used in the ribosome and phage display steps can be used to increase the chance of finding binders to the external or internal surface of the transporters. Finally, the generation of sybodies is quicker compared to nanobody generation, requires less protein, and does not require animal handling.

Here, the inactivation of the population of right-side out oriented $Glt_{Tk}$ by sybody binding made it possible to analyze uptake catalyzed by inside-out oriented proteins. The aspartate transport rates obtained in the presence of the sybody can be compared directly with a previous pre-equilibrium binding study on detergent-solubilized $Glt_{Ph}$. In the latter study, the transporter was fixed in the inward-oriented state by $Hg^{2+}$-crosslinking of a double cysteine mutant, which allowed for the determination of the kinetic mechanism of binding of sodium ions and aspartate, and estimation of rate constants for association and dissociation[36]. In our study, where we used a sybody to inactivate the population of right-side-out oriented $Glt_{Tk}$ transporters in proteoliposomes, we measured the kinetics of the reversed transport step, which includes binding of $Na^+$ and aspartate to the inward-oriented state similar to the pre-equilibrium binding study. In the $Na^+$ concentration range between 5 and 300 mM, the two studies are fully congruent and consistent with a mechanism in which two sodium ions bind first, followed by aspartate, and finally the last sodium ion. The kinetic analysis presented here shows that the rate constant for association of aspartate ($k_3$ in Eq. (1)), can be obtained using Eq. (13) when $K_M^{Asp}$ (App) is determined in the limit of high $Na^+$ concentrations (as shown in Fig. 2) and the turnover number $k_{cat}$ is taken directly from the maximally attainable rate at high $Na^+$ and aspartate concentrations ($v_{max}$). The value for $k_3$ derived in this way was $1.3 \times 10^6$ M$^{-1}$ s$^{-1}$, which closely matches the value of $1.2 \times 10^6$ M$^{-1}$ s$^{-1}$ for $Glt_{Ph}$ derived from pre-equilibrium binding experiments[36].

More importantly, using the same analysis, it is also possible to determine the turnover number $k_{cat}$, if $k_3$ is known from binding experiments (which is the case for $Glt_{Ph}$). This notion is relevant, because quantification of the amount of active transport protein present in membranes is often difficult, making a direct determination of $k_{cat}$ from $v_{max}$ values notoriously error-prone, which can be easily illustrated by the available data on $Glt_{Ph}$. For aspartate transport by $Glt_{Ph}$, the value for $K_M^{Asp}$ (app) at 100 mM $Na^+$ has been determined accurately (120 nM)[43]. Assuming that this $Na^+$ concentration is sufficiently high to represent the limit where $K_M^{Asp}$ (app) has become constant and using the $k_3$ value of $1.2 \times 10^6$ M$^{-1}$ s$^{-1}$ determined by pre-equilibrium binding, from Eq. (13) a $k_{cat}$ value of 0.14 s$^{-1}$ is calculated. This value has been derived without using the protein concentration.

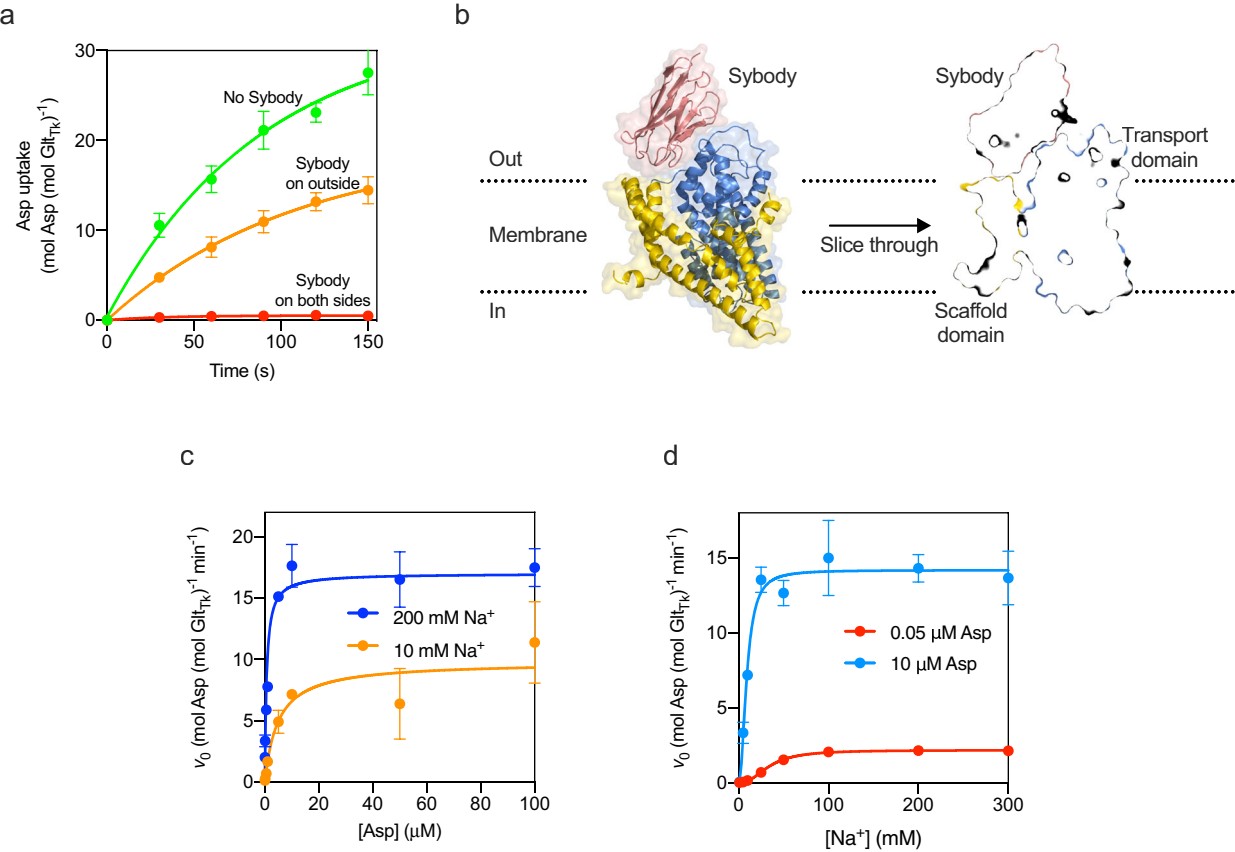

**Fig. 4 Inhibition of aspartate transport by the sybody. a** Uptake of aspartate in liposomes reconstituted with $Glt_{Tk}$ using aspartate and $Na^+$ concentrations of 1 µM and 300 mM respectively. Uptake traces in the absence of sybody (green), in the presence of 750 nM sybody on the outside (orange), or with sybody present on both sides of the membrane (red). Each data point represents a triplicate measurement ($n = 3$), and the standard error of the mean is shown. **b** Cryo-EM structure of the $Glt_{Tk}$-sybody complex. Left: cartoon representation of the $Glt_{Tk}$ protomer (transport domain in blue and the scaffold in yellow) that has the sybody bound (red). The surface of the protein is shown in transparent representation, and the approximate location of the membrane boundaries is indicated with dashed lines. Right: sliced-through representation highlighting that the sybody likely blocks movement of the transport domain along with the scaffold domain. The transport domain is in the intermediate-outward state, as described in ref. [13] **c, d** Same as in Fig. 1a, b for a selection of conditions (using the same color coding), but now in the presence of 750 nM sybody on the outside of the liposomes.

## Table 5 Cryo-EM data collection, refinement, and validation.

| | |
|---|---|
| Data collection and processing | |
| Voltage (kV) | 200 |
| Electron exposure (e⁻/Å²) | 53.3 |
| Defocus range (µm) | −0.5 to −2.0 |
| Pixel size (Å) | 1.012 |
| Symmetry imposed | C1 |
| Initial dataset (# of particles) | 109217 |
| Final dataset (# of particles) | 53983 |
| Map resolution (Å) FSC₀.₁₄₃ | 3.5 |
| Refinement | |
| Initial model used | PDB 6XWQ |
| Model composition | |
| Nonhydrogen atoms | 10357 |
| Protein residues | 1378 |
| Ligands | 3 |
| Mean $B$ factors (Å²) | |
| Protein | 160.2 |
| Ligand | 165.8 |
| Rms deviations | |
| Bond lengths (Å) | 0.005 |
| Bond angles (°) | 0.635 |
| Validation | |
| MolProbity score | 2.09 |
| Clash score | 17.7 |
| Poor rotamers (%) | 0.00 |
| Ramachandran plot (%) | |
| Favored | 95.07 |
| Allowed | 4.93 |
| Outliers | 0.00 |
| Model to map fit CC | 0.86 |

For comparison, if $k_{cat}$ is derived directly from the $v_{max}$ value of $3.4 \, \text{nmol} \times \text{mg}^{-1} \times \text{min}^{-1}$, a value of $2.6 \times 10^{-3} \, \text{s}^{-1}$ is found[43]. The huge discrepancy between the two values is probably caused by inaccurate protein concentration determination in the proteoliposomes, or loss of activity during the reconstitution process.

It is noteworthy that in the experiments on $Glt_{Tk}$ presented here, the $k_3$ value of $1.3 \times 10^6 \, \text{M}^{-1} \, \text{s}^{-1}$ that we determined from the measurement of $K_M^{Asp}$ (app) and $k_{cat}$, was remarkably similar to the experimentally determined value of $k_3$ for $Glt_{Ph}$ of $1.2 \times 10^6 \, \text{M}^{-1} \, \text{s}^{-1}$ [36]. The virtually identical rates of aspartate association step in $Glt_{Ph}$ and $Glt_{Tk}$ are consistent with the structures of the binding sites in the two proteins, which are essentially the same[7–9,13,15,16,20,44]. Therefore, we believe that the $Glt_{Tk}$ protein concentration, used to derive $k_{cat}$, was reasonably accurate in this case.

The sixfold difference in turnover number $k_{cat}$ between $Glt_{Ph}$ and $Glt_{Tk}$ ($0.14 \, \text{s}^{-1}$ and $0.9 \, \text{s}^{-1}$ respectively) could be caused by (small) structural differences away from the binding site. $k_{cat}$ is not a single rate constant but is composed of contributions from all steps that take place after binding of the last sodium ion in the catalytic cycle until the binding of the first sodium ion in the next round of catalysis. These steps include movement of the fully-loaded transport domain between outward- and inward-oriented states, the opening of the binding site towards the lumen of the liposomes, the release of the sodium ions and aspartate, occlusion

of the empty binding site in the apo-state, movement of the transport domain in the occluded apo-state, the opening of the binding site externally. The latter step also includes the rate constant for reshaping the binding sites for the first two sodium ions, leading to conformational selection that was shown to occur in $Glt_{Ph}$[34]. Therefore, differences between $Glt_{Ph}$ and $Glt_{Tk}$ that affect any of these steps may affect $k_{cat}$, which is observable in the value for $K_M^{Asp}$ (app) at high sodium concentrations.

The origin of the differences between $Glt_{Ph}$ and $Glt_{Tk}$ may be similar to that of differences between wild-type $Glt_{Ph}$ and a faster "unlocked" mutant[18,43]. This mutant has the same value for $k_3$ as the wild-type[36], consistent with the observation that the structure of the binding site is identical to the wild-type. Therefore, determination of $K_M^{Asp}$ (app) in the limit of high sodium ion concentration allows for the derivation of accurate values for $k_{cat}$. The values for $K_M^{Asp}$ (app) have been determined at 100 mM Na$^+$ for both wild-type $Glt_{Ph}$ and the fast mutant (120 nM and 406 nM respectively)[43]. If we again assume that this Na$^+$ concentration is high enough to represent the limit where $K_M^{Asp}$ (app) becomes constant for both proteins, then $k_{cat}$ is predicted to be ~3.5 times higher in the fast mutant, which is in good agreement with bulk transport data that showed ~4.5 fold difference in $v_{max}$ values (although the absolute numbers are both far off for the reasons discussed above)[43].

It is also possible to determine turnover numbers using a recently established transport assay at the single-molecule level[45], but combining turnover numbers from single-molecule transport measurements with rate constants determined in pre-equilibrium binding experiments is not straightforward. In pre-equilibrium binding experiments[33,34,36,37] and bulk transport experiments (presented here), the entire ensemble contributes to the measured rate constants, and therefore data can be combined to describe the ensemble properties. In contrast, in single-molecule transport experiments, the $k_{cat}$ is determined for only a fraction of the ensemble, and therefore it is difficult to quantitatively extract ensemble behavior, which would require accounting for the contribution of all subpopulations of (slightly) differently behaving individual proteins that together make up the ensemble[45]. Conversely, ensemble measurements also cannot predict the single-molecule behavior, but the observation that the $k_{cat}$ value determined by the ensemble measurements is lower than the maximal $k_{cat}$ value determined in single-molecule measurements is consistent with the heterogeneity at the single-molecule level.

While the kinetic mechanism presented here is valid only for the reverse transport reaction (because we inactivated the forward-operating transporters by the sybody), it is well possible that the same mechanism is also used for the forward transport reaction. The similarity of the forward and reverse kinetic mechanisms is indeed supported by the essentially identical kinetic parameters in the presence and absence of the sybody. From a structural point of view, this similarity can be explained because the transporter uses the same gating element on both sides of the membrane ("one-gate elevator"), and the binding site geometry and access path in the inward- and outward-oriented states are essentially the same[13,20,46,47]. In future experiments, the kinetic mechanism for forwarding transport by $Glt_{Tk}$ may be tested if we manage to inactivate the inside-out oriented population of $Glt_{Tk}$ molecules by a suitable sybody.

In contrast to what we found for $Glt_{Tk}$, differences in the kinetic mechanisms of the forward and reverse reactions have been reported for the mammalian glutamate transporter EAAC1[48]. The analysis of EAAC1 transport was based on the rapid-equilibrium assumption, which may be valid in this case, although it has not been tested systematically. Whether potential

mechanistic differences between the mammalian and archaeal transporters reflect evolutionary pressure to support different physiological needs is not clear. Alternatively, it is possible that the apparent differences in kinetic mechanism between $Glt_{Tk}$ and EAAC1 are caused by differences in the readout of transport. While transport was measured directly (using radioactivity) for $Glt_{Tk}$, for EAAC1 the substrate-induced chloride conductance was used as a readout. In addition, different ranges of co-ion concentrations were used in the study of EAAC1 compared to the work presented here, which also may make the two studies not directly comparable.

In conclusion, analysis of the kinetic mechanism of sodium-coupled aspartate transport by $Glt_{Tk}$ and $Glt_{Ph}$ provides a way to determine accurate turnover numbers from $K_M^{Asp}$ (app) values, without the need to use error-prone protein quantification. This is of great use for instance when analyzing the effects of mutations. Depending on the details of the kinetic mechanism, it may also be possible to determine turnover numbers in a similar way for other transport proteins. To test whether kinetic mechanisms different from the one analyzed here (Eq. (1)) would also allow for a similar determination of the turnover number, we used the King-Altman method[38,41] to derive rate equations for all possible kinetic mechanisms leading to the coupled transport of three sodium ions and aspartate[40]. It turns out that if at least one sodium ion binds after aspartate, $K_M^{Asp}$ (app) values in the limit of high sodium concentration equal the ratio between $k_{cat}$ and the on-rate for aspartate binding (as in Eq. (13)). It is also noteworthy that this even holds if multiple sodium ions bind randomly (for instance if steps 1 and 2 in Eq. (1) would be random). Therefore, it is likely that more transporters can be analyzed in the same way as presented here, and we conclude that systematic analysis of transport rates, and derivation of the rate equation, are essential steps in the elucidation of transport mechanisms.

## Methods

**$Glt_{Tk}$ purification and reconstitution in proteoliposomes.** $Glt_{Tk}$ was produced in *Escherichia coli* strain MC1061 with the L-arabinose inducible vector pBad24 as described in Arkhipova et al. [14]. The cells were grown in LB media supplemented with 100 mg/L ampicillin. The expression was induced by the addition of 0.05% L-arabinose when the culture reached 0.8 $OD_{600}$. Three hours after induction the cells were harvested by centrifugation (7000 RPM, 15′, 4 °C Beckman JLA 9.1000) and resuspended in ice-cold 20 mM Tris-HCl pH 8. The cells were lysed by means of a cell disruptor cooled to 4 °C and operated at 25 PSI. The lysate went through an intermediate centrifugation (7500 RPM, 20′, 4 °C, Beckman JA25.50) step to remove cell debris, the supernatant was finally ultracentrifuged (40000 RPM 150′, 4 °C, Beckman 50.1 Ti) and the pellet was resuspended in 20 mM Tris-HCl pH 8 before storing the membrane vesicles at −80°C.

The membrane vesicles were then added to solubilization buffer (50 mM Tris-HCl pH8, 300 mM KCl, 1% DDM), incubated for 45′ on a rocking platform at 4 °C, and finally centrifuged (55,000 RPM, 30′, 4 °C, Beckman MLA 55) to separate the insoluble fraction from the solubilized protein. The supernatant was supplemented with 15 mM imidazole pH 8 and with 0.5 mL of Ni-Sepharose slurry pre-equilibrated with 50 mM Tris-HCl pH 8, 300 mM KCl. After 1 h of incubation the mixture was loaded onto a Poly-Prep column and unbound protein was allowed to flow through. The column was washed with 20 column volumes of washing buffer (50 mM Tris-HCl, 300 mM KCl, 60 mM imidazole, 0.15% DM) and finally eluted in three fractions of 300, 800, and 400 μL respectively using elution buffer (50 mM Tris-HCl, 300 mM KCl, 500 mM imidazole, 0.15% DM). The second fraction was loaded onto a Superdex-200I gel filtration column equilibrated with 10 mM HEPES pH 8, 100 mM KCl and 0.15%DM. The final concentration of the purified protein was determined by measuring the absorbance at 280 nm ($Glt_{Tk}$ ε = 37,360).

The lipids used to reconstitute $Glt_{Tk}$ contained a 3:1 mixture of *E. coli* lipid polar extract and egg phosphatidylcholine (PC) (Avanti). Liposomes were homogenized by extruding 11 times through a 400 nm pore size polycarbonate filter and subsequently diluted to 5 mg/mL in 50 mM potassium phosphate buffer (pH 7.0). To allow the insertion of the protein into the bilayer, the lipids were destabilized by step-wise addition of 10% Triton X-100 while scattering was followed at a wavelength of 540 nm. The titration was stopped once the absorption signal decreased to about 60% of the maximum value reached. Purified protein was added at a protein:lipid ratio (w/w) of 1:1600. The protein-lipid mixture was incubated for 30′ at RT, and then the detergent was removed by addition of

BioBeads in three steps: First 15 mg/mL BioBeads were added followed by incubation for 60′ at 4 °C, then 19 mg/mL BioBeads were added followed by overnight incubation at 4 °C. Finally, 29 mg/mL BioBeads were added followed by 120′ incubation at 4 °C. BioBeads were then removed and the proteoliposomes were pelleted (80,000 RPM, 25′, 4 °C, Beckman MLA80) and resuspended in 50 mM potassium phosphate buffer (pH 7) to a final lipid concentration of 20 mg/ml. The proteoliposomes were subjected to three cycles of freeze-thawing using liquid nitrogen and stored until use.

**Sybody selection**. Sybodies were selected against two $Glt_{Tk}$ cysteine mutants (298 C and 367 C), which while biotinylated and immobilized during the selection procedures would make extracellular and intracellular epitopes accessible for binding, respectively. Selection was done in the presence of 50 μM L-aspartate, 150 mM NaCl, and 0.15% DM according to an established in vitro selection platform that included ribosome display, two rounds of phage display, and ELISA (Zimmermann et al.[39,42]). During ELISA, every single clone was analyzed for binding against $Glt_{Tk}$ in the presence and absence of L-aspartate. Sequencing of 48 ELISA positive hits resulted in 42 unique sybody sequences (20 for the 298 C mutant and 22 for the C367 mutant).

**Sybody expression and purification**. Each of 42 sybodies was expressed in *E. coli*, purified from the periplasm using $Ni^{2+}$-affinity chromatography, and analyzed by size exclusion chromatography (SEC). Based on the quality of the SEC profiles (absence of aggregates, no interactions with column material, high yield) we selected 33 purified sybodies (14 for the 298 C mutant and 19 for the 367 C mutant), which were tested for their ability to inhibit $Glt_{Tk}$ transport of aspartate in uptake assays. For large scale purification of inhibitory sybody 1, a preculture of *E. coli* MC1061 transformed with pSB_initSB1 was used to inoculate 50 mL of TB medium supplemented with 25 μg/ml chloramphenicol. The culture was grown for 2 h at 37 °C while shaking at 200 rpm, the temperature was then lowered to 22 °C and let grow until OD ~0.8. The expression was induced by adding L-arabinose to a final concentration of 0.02% and let express overnight at 22 °C while shaking. Cells were pelleted and resuspended in 5 mL periplasmic extraction buffer (20% sucrose (w/v) 50 mM Tris-HCl (pH 8.0), 0.5 mM EDTA and 0.5 μg/ml lysozyme) and incubated on ice for 30 min, after incubation 20 mL of TBS (20 mM Tris-HCl (pH 7.4) 150 mM NaCl) supplemented with 1 mM $MgCl_2$ were added. The lysate was centrifuged at 4000 g for 20 min and the supernatant was transferred in a tube containing 500 μL of Ni-sepharose pre-equilibrated in TBS and supplemented with imidazole to a final concentration of 15 mM. After 1 h incubation on a shaking platform, the solution containing the SyBody was applied to a polyprep gravity column and the unbound fraction was let flow through. The resin was then washed with 10 CV of TBS supplemented with 30 mM imidazole and eluted in 500 μl of TBS supplemented with 300 mM imidazole. The sybody solution was then passed through a NAP-10 column equilibrated with the internal uptake buffer (10 mM potassium phosphate buffer (pH 7), 300 mM KCl) and stored at −80 °C. sybody 1 was selected from the library created by the mutant 298 C.

**Transport assay**. The lumenal buffer in each proteoliposome preparation was changed to 10 mM potassium phosphate buffer (pH 7) and 300 mM KCl. For this, the proteoliposomes were first pelleted (80,000 RPM, 25′, 4 °C, Beckman MLA80) and then resuspended in the lumenal buffer. After three freeze-thaw cycles, the suspension was extruded 11 times through a polycarbonate filter with 400-nm pore size in order to obtain homogeneously sized unilamellar vesicles which were pelleted (80,000 RPM, 25′, 4 °C, Beckman TLA100.3) and resuspended to a final lipid concentration of 100 mg/mL.

To completely inhibit aspartate transport of $Glt_{Tk}$ by sybody 1 (on both sides of the membrane) the lumenal solution was supplemented with 150 μM of sybody 1. After performing 6 freeze-thaw cycles, and extrusion (11 times) through a polycarbonate filter with 400-nm pore size, the homogeneous solution was pelleted (80,000 RPM, 20′, 4 °C, Beckman TLA100.1) and resuspended to a final lipid concentration of 100 mg/mL in a solution containing 75 μM of sybody 1, 10 mM potassium phosphate buffer (pH 7) and 300 mM KCl.

To inhibit only the right-side-out fraction of $Glt_{Tk}$ a homogeneous solution of proteoliposomes was prepared as described above (with lumenal buffer devoid of sybody) and after pelleting by ultracentrifugation it was resuspended to a final lipid concentration of 100 mg/mL with a solution containing 75 μM of sybody 1, 10 mM potassium phosphate buffer (pH 7) and 300 mM KCl.

To start the transport the proteoliposome suspension was diluted 100 fold into external buffer while stirring. The external buffer contained 10 mM potassium phosphate buffer (pH 7), 3 μM valinomycin, 1–300 mM NaCl, 0.05–100 μM L-aspartate; to balance the osmotic strength with that of the lumenal solution, choline chloride was added (Table 1). In case the sybody was present, the 100-fold dilution resulted in a final external concentration of sybody of 750 nM.

After the indicated incubation period (20 s for the data in Figs. 1 and 4c, d), 2 mL of ice-cold quenching buffer (10 mM potassium phosphate (pH 7), 300 mM KCl) was added. The content of the tube was then poured onto a BA 45 nitrocellulose filter which was then washed with 2 mL of quenching buffer. The filters were finally dissolved in scintillation cocktail Ultima Gold (Perkin Elmer) and the β-decay from the radiolabeled substrate was counted. The time-point zero measurements for each condition was measured by pipetting the liposome

suspension on the side of a test tube containing 200 μL of reaction buffer and subsequently flushing them in the reaction buffer with 2 mL of ice-cold quenching buffer (10 mM potassium phosphate (pH 7), 300 mM KCl).

The value for each uptake rate represents the average and standard error of three independent biological replicates (different batches of expressed, purified, and reconstituted protein), each constituted by two or three technical replicates. The substrate-dependent uptake rates obtained at a fixed concentration of $Na^+$ were plotted as a function of L-aspartate, and the Michaelis-Menten equation was fitted to the data to obtain apparent $K_M$ ($K_M^{Asp}$ (app)) and $v_{max}$ ($v_{max}^{Asp}$ (app)) values for different $[Na^+]$. The co-ion-dependent uptake rates obtained at a fixed concentration of L-aspartate were plotted as a function of $Na^+$, and the Hill equation was fitted to the data for obtaining $K_M$ ($K_D^{Asp}$ (app)), $v_{max}$ ($v_{max}^{Na}$ (app)), and $n_{Hill}$ values for different [L-Asp]. The statistical analysis of the data was executed in GraphPad Prism 9.

**Single particle Cryo-EM**. The structure of $Glt_{Tk}$ in complex with sybody 1 (molar ratio 3:1) was determined using essentially the same protocol as described in Arkhipova et al.[13], in the presence of 300 mM $Na^+$ and 50 μM L-Asp. The purified complex at the concentration of 0.5–1 mg/ml was applied onto freshly glow-discharged Quantifoil grids (Au R1.2/1.3, 300 mesh) at 22 °C and 100% humidity and plunged-frozen in liquid ethane. The Cryo-EM data were collected using 200-keV Talos Arctica microscope (Thermo Fisher). Cryo-EM image processing was performed using cryoSPARC software[49].

In brief, 824 micrographs were selected for the processing after motion correction and CTF estimation. The template for particle picking was generated from 100 manually picked particles. Template-based picking identified 109,217 particles. Subsequent 2D classification reduced the number of particles to 67,498 and subsequently 53,983 particles were left in the selected ab initio class. Final non-uniform 3D refinement resulted in a 3.5 Å map (with C1 symmetry applied), which was sharpened using Autosharpen Map procedure in Phenix[50] and used for model building using Coot[51]. The refinement of the coordinates was performed in the realspace refine module of Phenix[52]. The data collection and refinement statistics are shown in Table 5. Visualization and structure interpretation was carried out in UCSF Chimera[53] and PyMol (Schrödinger, LLC).

**Statistics and reproducibility**. Each uptake rate represents the average of three independent biological replicates (separate purifications and reconstitutions), each constituted by two technical replicates, and the standard error of the mean is shown in the figures and tables.

**Reporting summary**. Further information on research design is available in the Nature Research Reporting Summary linked to this article.

## Data availability

Data supporting the findings of this manuscript are available from the corresponding authors upon reasonable request. A reporting summary for this Article is available as a Supplementary Information file. The source data underlying Figs. 1–4 and Tables 1–4 are provided as a Supplementary Data 1. The three-dimensional cryo-EM density map of the glutamate transporter homolog $Glt_{Tk}$ bound to the sybody has been deposited in the Electron Microscopy Data Bank under accession number EMD-12314 (https://www.ebi.ac.uk/pdbe/emdb/). Coordinates of the corresponding five models have been deposited in the Protein Data Bank under the accession number 7NGH (https://www.rcsb.org/).

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

## Acknowledgements

We thank Jan Rheinberger for help with cryo-EM sample preparation and data collection. This work was supported by the Dutch Research Council (NWO TOP grant 714.018.003 to DJS) and EMBO (short-term fellowship to AAG).

## Author contributions

G.T. performed all experiments with the exception of the sybody selection (done by A.A.G. with help and supervision from C.A.J.H. and M.A.S.) and the cryo-EM sample preparation (done by V.A.) and structure determination (V.A. and A.G.). All authors designed experiments and analyzed data. D.J.S. and G.T. wrote the manuscript with input from all other authors.

## Competing interests

The authors declare no competing interests.
