## [Peer Review File · Communications Biology]

Reviewers' comments:

Reviewer #1 (Remarks to the Author):

In this manuscript, Trinco et al. studies kinetic mechanisms of the prokaryotic Na⁺-coupled aspartate transporter GltTk. The authors determined aspartate uptake rates in proteoliposomes with purified transporters under various substrate concentrations and quantified transport rates with a rate equation for a generally assumed transport mechanism, in which two sodium ions bind before and another after aspartate. Since neurotransmitter transport might depend on the transport direction, protein orientation might be an important issue in reconstitution experiments. To overcome this limitation the authors developed a synthetic nanobody that blocks transporters from their outside and thus inactivates one population of reconstituted transporters. They demonstrate that oppositely oriented GltTk proteins exhibit the same transport rates, and explain this finding by the presence of identical gating element on both sides of the membrane. This is a very nice and important paper. I have only some comments.

1. Line 369: "This is especially problematic for GltPh, because this protein switches between active and quiescent modes, and may get stuck in kinetically distinct conformation by static disorder, all of which last for extended periods of time [46]." Transitions of active and quiescent modes have been described for many transporters, and thus far, there are only very limited methods to quantitatively describe these processes. The method described in this manuscript does not account for it, although high likelihoods of quiescent periods might result in a dramatic reduction in observed transport rates- The authors should mention this limitation and, if possible, might even speculate about potential ways to overcome this restrictions.

2. Line 375: "While the kinetic mechanism presented here is valid only for the reverse transport reaction ...it is well possible that the same mechanism is also used for the forward transport reaction." I do not understand why the authors did not use sybodies inside the vesicles for these experiments. Since they performed experiments with sybodies on both sides and with only external sybodies, this must be technically possible.

3. My last comment is just a curiosity. Can the sybody also bind to the transporter in the inward-facing conformation? How stable is binding, i.e. the authors assume an off-rate close to zero, is this a safe assumption? And if not, would this affect the validity of the conclusions?

Reviewer #2 (Remarks to the Author):

The manuscript titled "Kinetic mechanism of Na-coupled aspartate transport catalyzed by Glttk" by Slotboom et al reports a study of an aspartate/Na symporter Glttk. The authors measured the rate of aspartate uptake under different concentrations of Na, and constructed rates versus aspartate concentrations profiles for each Na concentration. Different kinetic schemes have to be employed to fit the profiles at different concentrations of Na, but overall the data seem consistent with a sequential binding-transport model in which two Na ion bind first, then the aspartate and then the third Na ion. The authors also developed a nanobody that blocks aspartate transport from one side of the transporter, and showed that the rate of aspartate transport is similar from either side of the transporter. Overall, the study is worthy of publication because this is certainly one of a kind in terms of systematically measuring aspartate transport in different combinations of Na and aspartate concentrations, and in demonstrating that the rate in either directions are similar. However, it is not clear how much of the kinetic schemes can be used to consolidate a single physical mechanism of transport or how different kinetic states corresponds to any of the observed structures. It is also not clear what structural features of the Glttk allow it to have similar rates of transport from inside to outside and the other way around, and whether this is generally applicable to other transporters in the same family.

Reviewer #3 (Remarks to the Author):

This is a well-conceived, quantitative study that details the kinetic parameters for the aspartate transporter GltTk in the forward and reverse transport modes. In particular, the interdependence of the co-transported substrates aspartate and Na⁺ are analyzed, allowing information about the

sequential nature of binding in this multi-substrate transporter. Importantly, proposals of binding sequential mechanisms from previous reports are confirmed, and aspartate binding rate and turnover number in the reverse direction are calculated. This was possible by selecting the reverse transport mode by inactivating the right-side-out transporter population with a sybody, an elegant method. Overall, this is a nice study based on high-quality experimental data. However, I have some minor points the authors might want to address:

1) From the sequence of the data presentation, it is not entirely clear which data reflect inside-out population only, and which data are for mixed populations. My guess is that all figures up to Fig. 4 are from mixed populations, but this should be made more clear.

2) The conclusion that pre-equilibrium conditions do not imply is important. However, it is not clear whether this pertains to aspartate binding only, or also to Na^+ ? it is likely that aspartate binding is not in pre-equilibrium at low concentrations, at which it most likely becomes rate limiting. Is this the case?

3) In the Zhang et al, PNAS 2007 report, pre-equilibrium conditions were used to model EAAC1 reverse transport kinetics. However, the binding rate constant for glutamate from the inside was determined as $10^7 \text{ M}^{-1}\text{s}^{-1}$, a value about 10-times higher than the one in this report. Together with the higher glutamate concentrations, this makes it less likely that pre-equilibrium conditions are invalid in the Zhang et al. report.

4) In mammalian glutamate transporters, it was suggested that the 100-fold lower affinity for glutamate binding to the inside-facing binding site, relative to the outside, is important for rapid dissociation of glutamate to the cytoplasm, in the background of high intracellular glutamate concentrations. What are the physiological concentrations of aspartate in the GltTk native bacterial environment? Could there be an evolutionary reason why asymmetry in the forward and reverse direction has evolved in one (mammalian), but not the other (bacterial, archaeal)?

5) Line 135: What is the reason for the claim that K_{mNa} (app) is not informative?

Reviewers' comments:

Reviewer #1 (Remarks to the Author):

In this manuscript, Trinco et al. studies kinetic mechanisms of the prokaryotic Na⁺-coupled aspartate transporter GltTk. The authors determined aspartate uptake rates in proteoliposomes with purified transporters under various substrate concentrations and quantified transport rates with a rate equation for a generally assumed transport mechanism, in which two sodium ions bind before and another after aspartate. Since neurotransmitter transport might depend on the transport direction, protein orientation might be an important issue in reconstitution experiments. To overcome this limitation the authors developed a synthetic nanobody that blocks transporters from their outside and thus inactivates one population of reconstituted transporters. They demonstrate that oppositely oriented GltTk proteins exhibit the same transport rates, and explain this finding by the presence of identical gating element on both sides of the membrane.

This is a very nice and important paper. I have only some comments.

1. Line 369: "This is especially problematic for GltPh, because this protein switches between active and quiescent modes, and may get stuck in kinetically distinct conformation by static disorder, all of which last for extended periods of time [46]." Transitions of active and quiescent modes have been described for many transporters, and thus far, there are only very limited methods to quantitatively describe these processes. The method described in this manuscript does not account for it, although high likelihoods of quiescent periods might result in a dramatic reduction in observed transport rates- The authors should mention this limitation and, if possible, might even speculate about potential ways to overcome this restrictions.

>We fully agree with the reviewer, and it was never our intention to claim that our method can quantitatively account for the switches between silent and active modes. We have rewritten the paragraph to avoid any misunderstanding (lines 371-384):

"It is also possible to determine turnover numbers using a recently established transport assay at the single molecule level [46], but combining turnover numbers from single molecule

transport measurements with rate constants determined in pre-equilibrium binding experiments, is not straightforward. In pre-equilibrium binding experiments [34,35,37,38] and bulk transport experiments (presented here), the entire ensemble contributes to the measured rate constants, and therefore data can be combined to describe the ensemble properties. In contrast, in single molecule transport experiments, the k_{cat} is determined for only a fraction of the ensemble, and therefore it is difficult to quantitatively extract ensemble behaviour, which would require accounting for the contribution of all subpopulations of (slightly) differently behaving individual proteins that together make up the ensemble [46]. Conversely, ensemble measurements also cannot predict the single molecule behaviour, but the observation that the k_{cat} value determined by the ensemble measurements is lower than the maximal k_{cat} value determined in single molecule measurements is consistent with the heterogeneity at the single molecule level.”

2. Line 375: “While the kinetic mechanism presented here is valid only for the reverse transport reaction ...it is well possible that the same mechanism is also used for the forward transport reaction.” I do not understand why the authors did not use sybodies inside the vesicles for these experiments. Since they performed experiments with sybodies on both sides and with only external sybodies, this must be technically possible.

> It is technically not possible to include the nanobodies only in the lumen. The procedure to incorporate the nanobodies in the liposome lumen includes an extrusion step in which the nanobody is presented to both sides of the membrane, which will lead to binding to all Glt_{TK} molecules present in the sample. To subsequently remove the external nanobodies would require extensive dialysis. Even if it was possible to quantitatively remove all the external nanobodies, the lengthy dialysis procedure leads to renewed liposome remodeling, including fusion events, as well as to protein inactivation, both of which are difficult to account for. We now explicitly mention this point in the manuscript (lines 267-271):

“It is important to note that the procedure to load sybodies in the liposome lumen includes an extrusion step in which all Glt_{TK} molecules in the sample are exposed to the sybody.

Therefore, it is not possible to do the opposite experiment, with the sybody exclusively included in the liposome lumen.”

3. My last comment is just a curiosity. Can the sybody also bind to the transporter in the inward-facing conformation?

> The sybody epitope is located in part on the scaffold domain, and in part on the transport domain (Figure 4), removing part of the interaction surface (by moving the transport domain to the inward-facing state) disrupts the epitope.

How stable is binding, i.e. the authors assume an off-rate close to zero, is this a safe assumption? And if not, would this affect the validity of the conclusions?

> We do not know the off-rates, but from the result presented in figure 4A where the sybody is present on both sides of the liposome, we can safely conclude that the concentration of sybody used is sufficient to fully block transport at least at the timescales of the experiments. If the fraction of sybody-free Glt_{TK} was significant, we would see accumulation of radiolabeled aspartate over time. The absence of accumulation justifies the interpretation of the experiments in which the sybody (at the same concentration) was added only on the outside.

Reviewer #2 (Remarks to the Author):

The manuscript titled “Kinetic mechanism of Na-coupled aspartate transport catalyzed by GlttK” by Slotboom et al reports a study of an aspartate/Na symporter GlttK. The authors measured the rate of aspartate uptake under different concentrations of Na, and constructed rates versus aspartate concentrations profiles for each Na concentration. Different kinetic schemes have to be employed to fit the profiles at different concentrations of Na, but overall the data seem consistent with a sequential binding-transport model in which two Na ion bind first, then the aspartate and then the third Na ion. The authors also developed a nanobody that blocks aspartate transport from one side of the transporter, and showed that the rate of aspartate transport is similar from either side of the transporter. Overall, the study is worthy of publication because this is certainly one of a kind in terms of systematically measuring aspartate transport in different combinations of Na and aspartate concentrations, and in demonstrating that the rate in either directions are similar. However, it is not clear how much of the kinetic schemes can be

used to consolidate a single physical mechanism of transport or how different kinetic states corresponds to any of the observed structures.

> In general, it is very difficult to unequivocally assign structural to kinetic states. However, we have indicated that the structures are fully consistent with the kinetic mechanism (lines 66-75):

“Structures of Glt_{Ph} and Glt_{Tk} have provided a qualitative explanation for the observed binding order. Two of the sodium binding sites (named Na1 and Na3) are buried deep in the proteins [15]. A substantial conformation rearrangement in the apo-protein (most pronounced in the conserved unwound region of TM7) is required to create the geometry needed for sodium binding, which makes this step slow. The conformational rearrangement, which is stabilized by binding of the two sodium ions, also affects residues involved in aspartate binding. While the apo state does not have measurable affinity for aspartate, sodium binding creates a high-affinity site for amino acid substrate. The last sodium ion binds to a site in a direct contact with HP2 gate (Na2), and locks the gate in the closed position, with aspartate and the three sodium ions occluded from the environment.”

It is also not clear what structural features of the Glttk allow it to have similar rates of transport from inside to outside and the other way around, and whether this is generally applicable to other transporters in the same family.

> We provide the information on GltTk in lines 389-392. Whether it is generally applicable to the entire family is unknown (see our response to reviewer 3)

“From a structural point of view this similarity can be explained because the transporter uses the same gating element on both sides of the membrane (“one-gate elevator”), and the binding site geometry and access path in the inward- and outward-oriented states are essentially the same [13,21,47,48].”

Reviewer #3 (Remarks to the Author):

This is a well-conceived, quantitative study that details the kinetic parameters for the aspartate transporter GltTk in the forward and reverse transport modes. In particular, the interdependence of the co-transported substrates aspartate and Na⁺ are analyzed, allowing information about the sequential nature of binding in this multi-substrate transporter. Importantly, proposals of binding sequential mechanisms from previous reports are confirmed, and aspartate binding rate and turnover number in the reverse direction are calculated. This was possible by selecting the reverse transport mode by inactivating the right-side-out transporter population with a sybody, an elegant method. Overall, this is a nice study based on high-quality experimental data. However, I have some minor points the authors

might want to address:

1) From the sequence of the data presentation, it is not entirely clear which data reflect inside-out population only, and which data are for mixed populations. My guess is that all figures up to Fig. 4 are from mixed populations, but this should be made more clear.

> The reviewer is correct in assuming that figures 1-3 represent data with mixed populations of transporters. We have now added this information to the legends

2) The conclusion that pre-equilibrium conditions do not imply is important. However, it is not clear whether this pertains to aspartate binding only, or also to Na⁺? it is likely that aspartate binding is not in pre-equilibrium at low concentrations, at which it most likely becomes rate limiting. Is this the case?

> The reviewer is correct that Na⁺ binding may be near equilibrium during the transport experiments (see Oh and Boudker 2018), but since we cannot measure Na⁺ transport directly, this notion does not help us in elucidation the kinetic mechanism. The reviewer is also correct that at very low Na⁺ concentrations the rapid equilibrium assumption may hold. But also this notion is not of practical use, because the sensitivity of the uptake assay is not sufficient to detect aspartate transport at very low sodium ion concentrations. We now mention this point in the manuscript (lines 162-164):

“It is noteworthy that the rapid equilibrium assumption might hold at very low Na⁺ concentration [37], but as discussed above, the sensitivity of the radiolabel-based transport assays is not high enough to measure aspartate uptake in such conditions.”

3) In the Zhang et al, PNAS 2007 report, pre-equilibrium conditions were used to model EAAC1 reverse transport kinetics. However, the binding rate constant for glutamate from the inside was determined as $10^7 \text{ M}^{-1}\text{s}^{-1}$, a value about 10-times higher than the one in this report. Together with the higher glutamate concentrations, this makes it less likely that pre-equilibrium conditions are invalid in the Zhang et al. report.

> We agree with this possibility and have rephrased the paragraph (lines 395-398).

“In contrast to what we found for Glt_{TK}, differences in the kinetic mechanisms of the forward and reverse reactions have been reported for the mammalian glutamate transporter EAAC1 [49]. The analysis of EAAC1 transport was based on the rapid-equilibrium assumption, which may be valid in this case, although it has not been tested systematically.”

4) In mammalian glutamate transporters, it was suggested that the 100-fold lower affinity for glutamate binding to the inside-facing binding site, relative to the outside, is important for rapid dissociation of glutamate to the cytoplasm, in the background of high intracellular glutamate concentrations. What are the physiological concentrations of aspartate in the GltTk native bacterial environment? Could there be an evolutionary reason why asymmetry in the forward and reverse direction has evolved in one (mammalian), but not the other (bacterial, archaeal)?

> Unfortunately, very little is known about the physiology of *Thermococcus kodakarensis*. We added a sentence to clarify (lines 398-400):

“Whether potential mechanistic differences between the mammalian and archaeal transporters reflect evolutionary pressure to support different physiological needs is not clear.”

5) Line 135: What is the reason for the claim that K_{MNa} (app) is not informative?

> We have added a clarification (lines 135-137):

“ K_{MNa} (app) and n_{Hill} (app) contain less useful information for discrimination between different mechanisms (as discussed in references [39,41]), and therefore these values were not used further here.”